# OpenReview forum: "Evaluating Gender Bias Transfer between Pre-trained and Prompt-Adapted Language Models"
_ICLR.cc/2025/Conference — Submitted to ICLR 2025_

### Official Review · Reviewer_XDWp · 2024-11-03

**Soundness:** 3
**Presentation:** 2
**Contribution:** 3
**Rating:** 8
**Confidence:** 4

**Summary:**

The paper investigates how gender biases in pre-trained large language models (LLMs) transfer to their adapted forms, particularly when using prompting methods like zero- and few-shot prompting instead of full fine-tuning. The study assesses the bias transfer hypothesis (BTH), which suggests that social biases embedded in pre-trained models persist and influence behavior in adapted tasks. The study underscores the necessity of addressing biases during the pre-training of language models to minimize downstream impacts in real-world applications.

**Strengths:**

1. This paper presents a substantive and meaningful contribution in examining bias transfer in prompt-adapted language models, filling a notable gap in current literature by focusing on causal models and prompting rather than only on fine-tuned or masked language models.

2. The quality of the paper’s methodology and experiments is strong.

**Weaknesses:**

1. While the study employs zero-shot, few-shot, and pre-prompting methods to analyze bias transfer, it does not explore other potentially impactful prompting techniques, such as chain-of-thought prompting, which has shown to affect LLM responses in reasoning tasks. Additionally, the prompts used are static, but dynamic or adaptive prompting—where prompts evolve based on intermediate model responses—could offer additional insight into the biases that emerge as models adapt to changing inputs. Including a broader variety of prompting techniques would increase the study's comprehensiveness and potentially reveal nuanced aspects of bias transfer.

2. A key weakness lies in the exclusive focus on gender bias within a binary framework. Although gender bias is a pressing issue, expanding bias analysis beyond binary gender categories would add substantial depth to the study. Including gender-neutral or non-binary pronouns, as well as biases related to other demographic dimensions (e.g., race, age, socioeconomic status), would reflect a more comprehensive understanding of model fairness.

3. The paper highlights how fairness-inducing and bias-inducing pre-prompts impact bias levels but does not deeply examine why certain prompts (e.g., negative or bias-inducing prompts) are more effective at altering bias than positive prompts. This leaves an important gap in understanding how pre-prompts function on a mechanistic level. A qualitative analysis of model outputs in response to different types of prompts, or a breakdown of how pre-prompts influence intermediate layers, could help clarify why some prompts succeed in reducing bias more than others.

**Questions:**

1. Have you considered testing for additional types of biases beyond binary gender (e.g., racial or intersectional biases)? If so, what challenges did you encounter that led you to focus solely on gender?

2. Why did you choose zero-shot, few-shot, and pre-prompting as the primary adaptation methods? Did you consider exploring other prompt strategies like chain-of-thought or dynamic prompting, which might have unique effects on bias transfer?

3. What specific barriers do you anticipate in implementing fairer pre-training, and how do you suggest practitioners address them?

---

> ### Author Response · Authors · 2024-11-16
>
> Reviewer comments
> ---
>
> W2. A key weakness lies in the exclusive focus on gender bias within a binary framework. Although gender bias is a pressing issue, expanding bias analysis beyond binary gender categories would add substantial depth to the study. Including gender-neutral or non-binary pronouns, as well as biases related to other demographic dimensions (e.g., race, age, socioeconomic status), would reflect a more comprehensive understanding of model fairness.
>
> Q1. Have you considered testing for additional types of biases beyond binary gender (e.g., racial or intersectional biases)? If so, what challenges did you encounter that led you to focus solely on gender?
>
> ---
> Response
> ---
>
> Fundamentally, we embrace the suggestion of analyzing other forms of bias and we are currently running experiments using the BBQ-lite dataset [1] to expand analysis to other demographic dimensions beyond gender. However, in the spirit of the importance of our findings, we argue that even demonstrating the transfer of a single (though significant) bias—binary gender—is valuable. While additional analyses on other bias dimensions or a more fine-grained definition of gender would certainly be beneficial, our results on binary gender transfer are already highly concerning. Key questions we pose are:  Would you say addressing binary gender bias transfer be a priority, even if other biases do not transfer? Is it important to raise awareness and incentivize the community to focus on mitigating this specific issue?
>
> That said, we did consider evaluating biases using non-binary genders, and intersectionality, but we decided to study bias transfer through the lens of binary gender due to the following reasons:
>
> * Evaluation of non-binary genders: As mentioned in the limitation section of the paper, we found that gender neutral datasets such as [2] and [3] introduce gender neutral variants of the WinoBias dataset, but we are unclear on when a “they / them'' pronoun in a sentence is a gender neutral singular reference vs plural reference. We believe meaningful analysis of biases against non-binary pronouns will require careful construction of unambiguously gender neutral fairness datasets, which we leave for future work as it is beyond the scope for the intended purposes of this paper. Are you aware of any other existing suitable dataset that uses non-binary gender and that we could adopt?
> * Evaluation of intersectional biases: Intersectionality is a hard bias problem — complexity of computational requirements, evaluation, analysis and interpretability of intersectional biases are much more challenging than single axis bias. While we agree with this comment and with the merit of identified opportunities, we believe intersectionality requires careful and nuanced research that deserves dedicated focus as follow up to our work. We plan to update the limitations section in the paper to call this out explicitly and recommend as future work.
>
> ---
> References
> ---
>
> [1] Parrish, A., Chen, A., Nangia, N., Padmakumar, V., Phang, J., Thompson, J., ... & Bowman, S. R. (2021). BBQ: A hand-built bias benchmark for question answering. arXiv preprint arXiv:2110.08193.
>
> [2] Dawkins, H. (2021). Second Order WinoBias (SoWinoBias) Test Set for Latent Gender Bias Detection in Coreference Resolution. arXiv preprint arXiv:2109.14047.
>
> [3] Vanmassenhove, E., Emmery, C., & Shterionov, D. (2021). Neutral rewriter: A rule-based and neural approach to automatic rewriting into gender-neutral alternatives. arXiv preprint arXiv:2109.06105.

---

> ### Author Response · Authors · 2024-11-16
>
> Reviewer comment
> --
>
> W3. The paper highlights how fairness-inducing and bias-inducing pre-prompts impact bias levels but does not deeply examine why certain prompts (e.g., negative or bias-inducing prompts) are more effective at altering bias than positive prompts. This leaves an important gap in understanding how pre-prompts function on a mechanistic level. A qualitative analysis of model outputs in response to different types of prompts, or a breakdown of how pre-prompts influence intermediate layers, could help clarify why some prompts succeed in reducing bias more than others.
>
> ---
> Response
> --
>
> We agree with that understanding how pre-prompting works on a mechanistic level, and a breakdown of how pre-prompts influence intermediate layers are rich areas of research that deserve dedicated research into why bias transfers. In order to avoid diluting the important message that bias does transfer in causal models with prompting, and due to space and time limits, we will explore this suggestion in future work. We will add this to a new future work section in the paper.

---

> ### Author Response · Authors · 2024-11-16
>
> Reviewer comment
> ----
>
> W1. While the study employs zero-shot, few-shot, and pre-prompting methods to analyze bias transfer, it does not explore other potentially impactful prompting techniques, such as chain-of-thought prompting, which has shown to affect LLM responses in reasoning tasks. Additionally, the prompts used are static, but dynamic or adaptive prompting—where prompts evolve based on intermediate model responses—could offer additional insight into the biases that emerge as models adapt to changing inputs. Including a broader variety of prompting techniques would increase the study's comprehensiveness and potentially reveal nuanced aspects of bias transfer.
>
> Q2. Why did you choose zero-shot, few-shot, and pre-prompting as the primary adaptation methods? Did you consider exploring other prompt strategies like chain-of-thought or dynamic prompting, which might have unique effects on bias transfer?
>
> ----
> Response
> ----
>
> Thank you for identifying the opportunity to clarify our experimental choices. In this work, after deliberation, the research team decided to work with prompting adaptation strategies that are simple enough to be adopted by typical non-expert LLM users; we believe this to be a key first step in better understanding impact to real world users. We will update our paper’s main body to transparently reflect the motivation behind our choice of prompting strategies for experiments.
>
> While it maybe out of scope for this work to comprehensively evaluate all prompting strategies available, as a result of this feedback:
>
> 1. we will expand analysis in this paper to Chain-of-Thought prompting (with the intent of including in the main paper in the upcoming week, which you rightly point out as having impact on model reasoning, and
> 2. we will explicitly encourage future work to expand to other more complex and nuanced prompting strategies.

---

> ### Author Response · Authors · 2024-11-16
>
> Reviewer comment
> ----
>
> Q3. What specific barriers do you anticipate in implementing fairer pre-training, and how do you suggest practitioners address them?
>
> ----
> Response
> ----
>
> This is a really good question. We do have some thoughts that we will elaborate on below, but we do not have all the answers. We hope papers like this one will stimulate the research community to study these challenges and offer solutions. In the meantime, for practitioners who pre-train foundation models using large-scale web datasets, we suggest to adopt careful data collection / curation strategies to limit representational harms for under-represented demographics. We encourage practitioners to refer to a growing body of dedicated literature [1, 2] that discuss challenges, implications and recommendations of fair curation of large pre-training datasets.
>
> We believe our work supports the need for more targeted research into creating fairer foundation models. Do you think the main paper would benefit from inclusion of a summary of above analysis?
>
>
> ----
> References
> ----
>
> [1] Zhao, D., Scheuerman, M. K., Chitre, P., Andrews, J. T., Panagiotidou, G., Walker, S., ... & Xiang, A. (2024). A Taxonomy of Challenges to Curating Fair Datasets. arXiv preprint arXiv:2406.06407.
>
> [2] Anthis, J., Lum, K., Ekstrand, M., Feller, A., D'Amour, A., & Tan, C. (2024). The Impossibility of Fair LLMs. arXiv preprint arXiv:2406.03198.

---

> ### Comment · Reviewer_XDWp · 2024-11-16
>
> Thank you for your detailed response. It addressed my concerns.I have therefore raised my rating.

---

> > ### Author Response · Authors · 2024-11-29
> > **Updated PDF and new experimental results**
> >
> > We thank you for your thoughtful feedback, which helped clarify and strengthen our paper.
> >
> > Specifically:
> >
> > 1. We uploaded a new version of PDF with clarifications highlighted in blue and red.
> >
> > 2. We conducted additional experiments to offer additional empirical analysis that reveal that **bias does transfer in causal models across different demographics (such as age, nationality, socio-economic status and nationality), and for CoT prompting**. These findings significantly amplify the impact of our research, extending its relevance and implications beyond just binary-gender, and traditional zero- and few-shot prompting paradigms. We included our initial results in our detailed comment above in the open review forum, and plan to complete and update the camera ready version of the paper if our work is accepted to ICLR.
> >
> > We kindly request you to consider the provided clarifications and new experimental findings in your assessment of our work.

---

### Official Review · Reviewer_1AiY · 2024-11-04

**Soundness:** 3
**Presentation:** 1
**Contribution:** 2
**Rating:** 5
**Confidence:** 4

**Summary:**

The paper studies the gender bias of autoregressive LLMs across several experimental settings while performing a coreference resolution task, focusing on *bias transfer*, which is the correlation between models' "intrinsic" bias (next-token prediction) versus their bias in zero- or few-shot prompting. The results indicate that standard prompt engineering techniques, such as modifying task instruction prompts or including few-shot task examples, have only a small effect on bias transfer.

**Strengths:**

The paper is generally well-written, with detailed and well-documented experiments and analysis that all appear to be sound. Comprehensive experiments are performed across a range of LLM prompting settings and three key open-source models.

**Weaknesses:**

The primary weaknesses of the paper are (1) its failure to acknowledge closely related prior work, and (2) its lack of novelty when the missing prior work is taken into account. There is a large family of work studying the same questions as in this work [1-5] -- i.e., zero- and few-shot evaluation of causal (autoregressive) LLM bias; see [5] for a helpful survey -- but none of it is cited. Instead, the paper is written as if the only prior studies of LLM bias had been conducted in the context of masked language models or fine-tuned autoregressive models, artificially inflating the paper's contribution.

[1] Ganguli, D., Askell, A., Schiefer, N., Liao, T. I., Lukošiūtė, K., Chen, A., ... & Kaplan, J. (2023). The capacity for moral self-correction in large language models. arXiv preprint arXiv:2302.07459.

[2] Bai, X., Wang, A., Sucholutsky, I., & Griffiths, T. L. (2024). Measuring implicit bias in explicitly unbiased large language models. arXiv preprint arXiv:2402.04105.

[3] Lin, L., Wang, L., Guo, J., & Wong, K. F. (2024). Investigating Bias in LLM-Based Bias Detection: Disparities between LLMs and Human Perception. arXiv preprint arXiv:2403.14896.

[4] Huang, D., Bu, Q., Zhang, J., Xie, X., Chen, J., & Cui, H. (2023). Bias assessment and mitigation in llm-based code generation. arXiv preprint arXiv:2309.14345.

[5] Ranjan, R., Gupta, S., & Singh, S. N. (2024). A Comprehensive Survey of Bias in LLMs: Current Landscape and Future Directions. arXiv preprint arXiv:2409.16430.

One interesting fact is that one of the primary arguments made in this paper -- that prompt engineering techniques such as optimizing task instruction prompts or including few-shot examples are not broadly effective tools for mitigating LLM bias -- seems to contradict the findings of several other works that have studied the same question [3, 4] (neither of which are cited in the paper). This disconnect is the result of the paper's overwhelming focus on *bias transfer* (the Pearson correlation coefficient between bias in next-token prediction and bias in zero- or few-shot prompting) rather than absolute bias numbers: i.e., if bias is significantly reduced using prompt engineering but remains directionally the same as in the "intrinsic" setting, then bias transfer remains high despite the reduction of bias
- E.g., in Table 2, for unambiguous (Type 2) task instances, the LLM bias (A-SB) goes from 27.7% in the "intrinsic" setting to 9.47% using a zero-shot prompt under the "fair" instruction prompt, and down to 4.16% using the same task instruction prompt in a few-shot setting; but the bias transfer (Pearson correlation) is $\rho \geq 0.92$.

Instead of presenting the paper as if it is the first work to consider research questions that have already been extensively studied, it would be interesting to examine whether the bias transfer metric remains useful in understanding LLM bias by comparing and contrasting with the findings presented by prior work. Such a paper might retain some novelty in what has become a crowded research area -- though there does appear to be at least one other paper considering this question as well [6], it only considers bias for a summarization task in the context of fine-tuning approaches, meaning that the specific experimental findings of this work would remain novel.

[6] Ladhak, F., Durmus, E., Suzgun, M., Zhang, T., Jurafsky, D., McKeown, K., & Hashimoto, T. B. (2023, May). When do pre-training biases propagate to downstream tasks? a case study in text summarization. In Proceedings of the 17th Conference of the European Chapter of the Association for Computational Linguistics (pp. 3206-3219).

Finally, there are a few confusing or misleading statements that should be revised:
- "MLMs fare better [than causal/autoregressive LLMs] at text classification and sentence analysis" (line 49-50) -- what is the evidence to support this statement?
- "causal models are much larger than MLMs" (line 50-51) -- while this is true of the leading available open-source models of each architecture, this is simply because causal models have become much more popular, so they have recently been scaled much larger; but scale is not an inherent feature of either architecture.
    - The sentence continues "[...] and therefore possess a greater capacity to perpetuate societal biases (Bender et al., 2021)." However, Bender et al. (2021) does not argue that LLMs' capacity to perpetuate societal bias is correlated with model scale (or the causal/autoregressive architecture).

**Questions:**

Argumentation:
- In line 42-45, it is claimed that "the conclusion that bias does not transfer [...] has potentially dire implications for the fairness of task-specific models beyond MLMs, as it implies that the fairness of the pre-trained model does not matter." -— Why is this considered "dire for fairness"? If bias does not transfer, this would mean that it is easier to mitigate the inherent bias of LLMs as compared to a scenario where bias does transfer.

Experimental design:
- What samples is the Pearson correlation in Figure 3 and sections 4.1 and 4.2 computed over? Is it the O-SB values for each sample, each occupation, or each occupation-ambiguity combination?
- What are each of the points in the scatterplot visualized in Figure 3? Is it O-SB for each occupation individually, with each point as a different occupation (and all using the same prompt)?
- How many few-shot examples are included for the results in Table 2?

(UPDATE: score increased from 3 to 5 following rebuttal.)

---

> ### Author Response · Authors · 2024-11-16
>
> Reviewer comment
> ---
>
> W1. The primary weaknesses of the paper are (1) its failure to acknowledge closely related prior work, and (2) its lack of novelty when the missing prior work is taken into account. There is a large family of work studying the same questions as in this work [1-5] — i.e., zero- and few-shot evaluation of causal (autoregressive) LLM bias; see [5] for a helpful survey — but none of it is cited. Instead, the paper is written as if the only prior studies of LLM bias had been conducted in the context of masked language models or fine-tuned autoregressive models, artificially inflating the paper's contribution.
>
> [1] Ganguli, D., Askell, A., Schiefer, N., Liao, T. I., Lukošiūtė, K., Chen, A., ... & Kaplan, J. (2023). The capacity for moral self-correction in large language models. arXiv preprint arXiv:2302.07459.
>
> [2] Bai, X., Wang, A., Sucholutsky, I., & Griffiths, T. L. (2024). Measuring implicit bias in explicitly unbiased large language models. arXiv preprint arXiv:2402.04105.
>
> [3] Lin, L., Wang, L., Guo, J., & Wong, K. F. (2024). Investigating Bias in LLM-Based Bias Detection: Disparities between LLMs and Human Perception. arXiv preprint arXiv:2403.14896.
>
> [4] Huang, D., Bu, Q., Zhang, J., Xie, X., Chen, J., & Cui, H. (2023). Bias assessment and mitigation in llm-based code generation. arXiv preprint arXiv:2309.14345.
>
> [5] Ranjan, R., Gupta, S., & Singh, S. N. (2024). A Comprehensive Survey of Bias in LLMs: Current Landscape and Future Directions. arXiv preprint arXiv:2409.16430.
>
> ---
> Response
> ---
>
> Thank you for the opportunity to clarify and for pointing us to those references. In what follows, we are going to explain how [1, 3, 4, 5] are different from our work. We will also discuss [2] which is actually very close in spirit (albeit also different in its approach and focus). [2] happens to be an arxiv-only submission which we were not aware of, while relevant submissions that do not appear in conference/journal proceedings are excluded by the ICLR policy.
>
> The research question we study in this paper is “can bias transfer from pre-trained causal models when prompt adapted?” We do not claim that previous literature does not study causal models for intrinsic biases, nor that causal models have not been studied for biases under prompt adaptations. Our claim is that the relationship between intrinsic biases and prompt adapted biases in causal models is not well understood. As detailed in our related work section, previous works largely study bias transfer only in masked language models under fine-tuning. Expanding our understanding of bias transfer to causal models under prompting is a fundamental question that is critical to answer to encourage future work to understand causal relationship between prompting and model biases.
>
> Are there specific instances of text in the paper that do not accurately reflect the above research goal? We would like to invite you to share unclear or misleading sentences in the paper, which we will be more than happy to update to accurately reflect our research goal as detailed above.
>
> Furthermore, thank you for sharing the above 5 papers for our consideration. While [1, 3, 4, 5] study biases in causal models under various applications, they do not study the relationship between intrinsic and prompted biases, which is the key contribution of our paper. [5] does a comprehensive survey on works that study biases upon prompting, but does not discuss bias transfer. Although papers [1, 3, 4, 5] are not directly tackling bias transfer, we are happy to augment our related works section to include a general review of bias investigations in causal models.
>
> Thank you for sharing [2] work with us! Our method for sourcing papers for this work was to predominately explore peer-reviewed conference proceedings and journals. As [2] is an arxiv-only paper that is currently not available as part of any proceedings, we missed it; while it appears that this work was accepted at a NeurIPS workshop in Oct 2024, it is not part of the proceedings at this time. Even though ICLR citation policy states that we “may be excused for not knowing about papers not published in peer-reviewed conference proceedings or journals, which includes papers exclusively available on arXiv”, we are happy to cite this work as a contemporary work in our related work section. This paper, like ours, studies the relationship between intrinsic and extrinsic biases in causal models. However, this work differs from our work as their focus is on measuring bias transfer only in settings where the model gates / rejects responses in the downstream setup. We believe our work, together with [2] will play an important role in raising awareness and can enable future researchers to make meaningful progress in better understanding bias impacts of adaptation strategies. Given ICLR’s policy on contemporary work, this should not impact our paper’s novelty.

---

> > ### Author Response · Authors · 2024-11-16
> >
> > Reviewer comment
> > ---
> >
> > W3. Instead of presenting the paper as if it is the first work to consider research questions that have already been extensively studied, it would be interesting to examine whether the bias transfer metric remains useful in understanding LLM bias by comparing and contrasting with the findings presented by prior work. Such a paper might retain some novelty in what has become a crowded research area — though there does appear to be at least one other paper considering this question as well [6], it only considers bias for a summarization task in the context of fine-tuning approaches, meaning that the specific experimental findings of this work would remain novel.
> >
> > [6] Ladhak, F., Durmus, E., Suzgun, M., Zhang, T., Jurafsky, D., McKeown, K., & Hashimoto, T. B. (2023, May). When do pre-training biases propagate to downstream tasks? a case study in text summarization. In Proceedings of the 17th Conference of the European Chapter of the Association for Computational Linguistics (pp. 3206-3219).
> >
> > ---
> > Response
> > ---
> >
> > Thank you for the thoughtful suggestion. As clarified earlier in our response to your concern regarding W1, our research investigation into bias transfer in causal models under prompting has not previously been well studied, and our key finding is in contradiction to existing literature in bias transfer analysis (which primarily studies MLMs and fine-tuning adaptations). We believe our work amounts to a critical novel finding regarding the bias impacts of commonly used adaptations.
> >
> > We believe this finding, similar to those in the contemporary work in paper 2, will be foundational to increasing awareness for, and incentivize important future research into (1) bias impacts of various adaptation strategies, (2) mechanistic explanations for why bias transfers in causal models under prompting (encouraging more works similar to that in paper 6), now that we have established that bias does transfer, and (3) targeted mitigations for fairer pre-training of causal models that will be prompted to achieve downstream tasks. Given that our research uncovers a key novel finding that bias can transfer from pre-trained causal models to downstream tasks using prompting, we believe ICLR to be an appropriate venue for this research given its guidelines for significance of research: "We encourage reviewers to be broad in their definitions of originality and significance".

---

> ### Author Response · Authors · 2024-11-16
>
> Reviewer comment
> ---
>
> W2. One interesting fact is that one of the primary arguments made in this paper -- that prompt engineering techniques such as optimizing task instruction prompts or including few-shot examples are not broadly effective tools for mitigating LLM bias — seems to contradict the findings of several other works that have studied the same question [3, 4] (neither of which are cited in the paper). This disconnect is the result of the paper's overwhelming focus on bias transfer (the Pearson correlation coefficient between bias in next-token prediction and bias in zero- or few-shot prompting) rather than absolute bias numbers: i.e., if bias is significantly reduced using prompt engineering but remains directionally the same as in the "intrinsic" setting, then bias transfer remains high despite the reduction of bias
>
> E.g., in Table 2, for unambiguous (Type 2) task instances, the LLM bias (A-SB) goes from 27.7% in the "intrinsic" setting to 9.47% using a zero-shot prompt under the "fair" instruction prompt, and down to 4.16% using the same task instruction prompt in a few-shot setting; but the bias transfer (Pearson correlation) is .
>
> ---
> Response
> ---
>
> This is a really good point to discuss. Our focus on bias transfer is by design, to ensure we are addressing the important question of “can bias transfer from causal pre-trained models to prompted models?” We do agree that there is an opportunity to clarify certain aspects of Sec 4.2 that studies the impact of fairness and bias inducing pre-prompts on bias transfer. In response to this feedback, in addition to our existing analysis (in lines 310-335), we plan to address the following points:
>
> 1. Explicitly remark on the effectiveness of fairness and bias inducing pre-prompts, citing previous works that have shown pre-prompting as an effective bias mitigation strategy (including but not limited to [3] and [4]).
>
> 2. Qualitatively explain that while absolute bias numbers are important, so too are bias correlations. They both offer valuable but different insights into model biases. In Table 2, as you rightly observe, fairness improves with the use of fairness inducing prompts. While fairness improves in an absolute sense, the model still embodies the same directional gender biases for occupations, as captured by Pearson Correlation and detailed in lines 333-335. These metrics capture inherently different bias qualities, which we will make sure to update in our analysis in Sec 4.2. We also intend to explicitly call out that the metrics that we propose motivate researchers and developers to consider and test for bias in the pre-trained models they use for their end tasks, as these biases can manifest themselves in the prompt-adapted models.

---

> > ### Author Response · Authors · 2024-11-16
> >
> > Reviewer comment
> > ---
> >
> > W4. Finally, there are a few confusing or misleading statements that should be revised:
> >
> > "MLMs fare better [than causal/autoregressive LLMs] at text classification and sentence analysis" (line 49-50) — what is the evidence to support this statement?
> >
> > "causal models are much larger than MLMs" (line 50-51) — while this is true of the leading available open-source models of each architecture, this is simply because causal models have become much more popular, so they have recently been scaled much larger; but scale is not an inherent feature of either architecture.
> >
> > The sentence continues "[...] and therefore possess a greater capacity to perpetuate societal biases (Bender et al., 2021)." However, Bender et al. (2021) does not argue that LLMs' capacity to perpetuate societal bias is correlated with model scale (or the causal/autoregressive architecture).
> >
> > ---
> > Response
> > ---
> >
> > Thank you for pointing out points of confusion, we agree that there is opportunity to be more specific and clear here. We will clarify these statements in the main body of paper and support statements with evidence, as advised. We will provided an updated draft and will point out these modifications to you in the coming week.

---

> ### Author Response · Authors · 2024-11-16
>
> Reviewer comment
> ---
>
> Q1. In line 42-45, it is claimed that "the conclusion that bias does not transfer [...] has potentially dire implications for the fairness of task-specific models beyond MLMs, as it implies that the fairness of the pre-trained model does not matter." -— Why is this considered "dire for fairness"? If bias does not transfer, this would mean that it is easier to mitigate the inherent bias of LLMs as compared to a scenario where bias does transfer.
>
> ---
> Response
> ---
>
> Thank you for the opportunity to clarify. In saying “bias does not transfer” we specifically mean that previous works [1-4], find that intrinsic biases in pre-trained MLMs bear limited effect on downstream biases (after fine-tuning). Previous literature attributes this finding to models learning biases present in fine-tuning datasets, even if intrinsic biases in pre-trained models are mitigated; the direct implication of these findings in prior literature is that downstream biases will persist despite intrinsic bias mitigation (as bias does not transfer). This finding suggests that practitioners do not have to care about the bias in pre-trained models, and that it is sufficient to mitigate biases at the downstream stage; while this may be a valid conclusion for some settings (MLMs when fine-tuning with lots of data), they are not generalizable to other settings as proven in our paper (causal models under prompting). When previous findings are taken out of context of their specific experimental setup, there is risk of practitioners discounting intrinsic biases in models altogether as a result of previous literature, explaining our remark on dire implications for fairness. In contradiction, our work strongly highlights the users need to carefully consider biases in pre-trained models without discounting them.
>
>
> ---
> References
> ---
>
> [1] Goldfarb-Tarrant, S., Marchant, R., Sánchez, R. M., Pandya, M., & Lopez, A. (2020). Intrinsic bias metrics do not correlate with application bias. arXiv preprint arXiv:2012.15859.
>
> [2] Steed, R., Panda, S., Kobren, A., & Wick, M. (2022, May). Upstream mitigation is not all you need: Testing the bias transfer hypothesis in pre-trained language models. In Proceedings of the 60th Annual Meeting of the Association for Computational Linguistics (Volume 1: Long Papers) (pp. 3524-3542).
>
> [3] Kaneko, M., Bollegala, D., & Okazaki, N. (2022). Debiasing isn't enough!—On the Effectiveness of Debiasing MLMs and their Social Biases in Downstream Tasks. arXiv preprint arXiv:2210.02938.
>
> [4] Schröder, S., Schulz, A., Kenneweg, P., & Hammer, B. (2023). So Can We Use Intrinsic Bias Measures or Not?. In ICPRAM(pp. 403-410).

---

> ### Author Response · Authors · 2024-11-16
>
> Reviewer comment
> ---
>
> Q2. What samples is the Pearson correlation in Figure 3 and sections 4.1 and 4.2 computed over? Is it the O-SB values for each sample, each occupation, or each occupation-ambiguity combination?
>
> ---
> Response
> ---
>
> For each model and adaptation strategy, we compute O-SB values for 40 WinoBias occupations over 5 random seeds, producing 200 O-SB values per adaptation strategy. Then, we compute Pearson correlation between O-SB values for intrinsic setting and O-SB values for zero/few-shot setting. An intuitive interpretation of this correlation is whether the occupational gender biases are directionally aligned (i.e., do pro-stereotypical biases remain pro-stereotypical with and without prompting, and similarly anti-stereotypical biases). Upon reflection, this could have been explained clearer in the paper. We will update lines 228 - 230 to state the following:
>
> “Lastly, similar to Steed et al. (2022), bias transfer between two adaptations is computed as the Pearson correlation coefficient (ρ). Here we measure the correlation between O-SB values in intrinsic and prompt-based evaluations across all occupations and random seeds. In our setting, the Pearson correlation for a given model measures directional alignment of occupational gender stereotypes across occupations and random seeds.”
>
> We will also provide a deeper analysis of when bias reduces and correlation contracts, to help the reader understand what causes a high Pearson correlation when bias is reduced.
>
> ---
> Reviewer comment
> ---
>
> Q3. What are each of the points in the scatterplot visualized in Figure 3? Is it O-SB for each occupation individually, with each point as a different occupation (and all using the same prompt)?
>
> ---
> Response
> ---
>
> Good question, each point on the scatterplot in Figure 3 represents O-SB for given occupation, experimental run and model. In any correlation plot in the paper, the x-axis for a given point represents the O-SB for intrinsic analysis whereas the y-axis represents the O-SB when the model was prompted with either zero- (Fig 3. left) or few-shot (Fig 3. right) prompts. These are the inputs to our Pearson correlation metric. For example, one of the points on the left scatterplot in Figure 3 represents the O-SB for the doctor occupation, when the random seed is set to 9, the x-coordinate is the intrinsic O-SB from Llama 3 8B Instruct while the y-coordinate is from zero-shot prompting Llama 3 8B Instruct. Figure 3 is computed using prompts that fundamentally leverage the same set of WinoBias sentences, but formatted in different ways for intrinsic, zero- and few-shot settings as detailed in Fig 1. More specifically, the pre-prompt used for zero- and few-shot prompts is “neutral”: “Choose the right option for the question using the context below”, and we use few-shot prompts in the 3-shot setting. We will make this more clear in the figure description by adding the following to the Figure 3 description:
>
> “Each point on the scatterplot represents O-SB for a single occupation, random seed, and model. Altogether these results span 40 occupations, 5 random seeds, and 4 models.”
>
> ---
> Reviewer comment
> ---
>
> Q4. How many few-shot examples are included for the results in Table 2?
>
> ---
> Response
> ---
>
> Thank you for pointing out this opportunity to clarify, we will update the main paper with the following information.
>
> “We use 3-shot prompts in Table 2; one of which is an unambiguous sentence with a pro-stereotypical answer, another is an unambiguous sentence with an anti-stereotypical answer, and the third is an ambiguous sentence with “unknown” as the right answer.“

---

> ### Author Response · Authors · 2024-11-25
> **Eager to engage with you during rebuttal**
>
> Thanks you for taking the time to review our submission. We are eager to engage with you during the rebuttal. What are your thoughts on our response and is there anything else we can help clarify?
>
> Looking forward to hearing from you on our paper.

---

> ### Comment · Reviewer_1AiY · 2024-11-27
> **Response to rebuttal**
>
> Thank you for your comprehensive response! I have read the general response to all reviewers, your rebuttal, and your changes to the paper draft highlighted in blue. These have resolved most of my concerns, but one key point remains.
>
> From your response, my understanding is this: for a suitably narrow and specific formulation of your research question -- e.g., let's call this "under *the proposed bias transfer metric*, do we see bias transfer for *autoregressive* LLMs, even when *prompts are engineered to mitigate bias*?", which I will refer to as **SRQ** -- then, on reading your response and the updated related work section, I agree that your study of SRQ is indeed novel. (Thank you for updating your related work section on this point and highlighting your changes for easy reference.)
>
> However, for a broader formulation of the question -- e.g., "does bias learned by LLMs during pre-training or fine-tuning lead to biased LLM behaviors, even under various mitigation strategies?", which I will refer to as **GRQ** -- the fact remains that this question has already been extensively studied. Thus, my primary remaining concern is this: **Why is it important to study this question according to **SRQ**, rather than other versions of **GRQ**?**
>
> To answer this question, it is necessary to justify two points:
> 1. Why is the proposed bias transfer metric a better approach to understanding LLM bias versus the approaches of other work?
> 2. Why is it important to study bias transfer specifically in the context of prompt adaptation (rather than, e.g., fine-tuning)?
>
> I do not find the argument in response to W2 and W3 convincing on either point (particularly (1)). *I am open to improving my assessment if the authors are able to provide a convincing answer to this concern that addresses both points.*
>
> ---
>
> ### Minor points
>
> Briefly, 2 additional points that I would like to communicate to authors -- these will not impact my decision, but they are important to communicate anyway:
> - You provided a great answer to my question about "why is it considered 'dire for fairness' if bias does not transfer?", but I don't see this updated in the draft. I think this is fine for now, as it is unreasonable to expect every single point in your responses to make it into a revised draft during the discussion period. However, for your next full draft, I would highly recommend clarifying this point as you did in your response to me, as this is a very important and nuanced point that can be easily misunderstood from the current text.
> - By the way, when you use the term "contemporary work" in your rebuttal, I think the term you are looking for is "concurrent" or "contemporaneous". This was confusing on my initial read of your rebuttal; but if this is what you meant, then I agree with your point on [ICLR's policy](https://iclr.cc/Conferences/2025/FAQ#:~:text=Q%3A%20What%20constitutes%20concurrent/contemporaneous%20work%2C%20and%20what%20is%20the%20relevant%20policy%20regarding%20it%3F) for "concurrent/contemporaneous work"

---

### Official Review · Reviewer_WiC1 · 2024-11-04

**Soundness:** 2
**Presentation:** 2
**Contribution:** 2
**Rating:** 5
**Confidence:** 2

**Summary:**

This paper shows how biases existing in pre-trained language models transfer to downstream tasks through prompt adaptation. It shows that there is a strong correlation between biases in pre-trained models and models with zero- and few-shot prompts. Even when models are specifically pre-prompted to promote fairness or bias, such a correlation is still very significant.

**Strengths:**

1. This paper explores bias transfer under various prompt settings, including zero-shot, few-shot, and pre-prompted configurations. It offers a thorough analysis of bias transfer for different prompting methods.

2. The strong correlation between intrinsic biases in pre-trained models and biases in prompt-adapted tasks emphasizes the need for fairness-aware pretraining or fairness-aware tuning.

**Weaknesses:**

1. The study primarily uses the WinoBias dataset for gender bias evaluation, and it ignores the biases related to other demographic factors, such as race.

2. The observations are interesting, but the explanations for the observations are not sufficient.

3. The novelty is quite limited. It neither does not provide deep explanations for the observations nor any methods to mitigate the bias.

**Questions:**

1. The observations are interesting, but they are kind of superficial for me. Could the authors provide any analysis to explain why the correlations exist? For example, some analysis is based on representations or logit lens.

2. If in-context learning (prompting) cannot be the promising solution for bias transfer. What is the next step? Could the authors talk more about the future work?

---

> ### Author Response · Authors · 2024-11-16
>
> Reviewer comment
> ---
>
> W1. The study primarily uses the WinoBias dataset for gender bias evaluation, and it ignores the biases related to other demographic factors, such as race.
>
> ---
> Response
> ---
> We agree that adding more analysis with other demographic axes could support our conclusions further, but we would argue that showing that one type of bias does transfer already validates our hypothesis that bias can transfer in causal models through prompting. We believe this awareness is crucial for both researchers and practitioners, even if only gender bias transfers (while all others do not). We welcome a discussion with you about the relevance of our findings, and what could happen with new findings on non-gender axes.
>
> In the meanwhile, while our work offers a focused discussion on gender stereotypes, we agree that expanding beyond gender to other demographic dimensions could help offer readers a holistic understanding of how biases transfer more generally. In response to this feedback, to supplement our existing results in section 4.1, we are currently in the process of running bias transfer evaluations on other axes such as age, race, disability, etc using the BBQ lite dataset [1]. We will update the paper with experimental results in the upcoming week.
>
>
> ---
> References
> ---
> [1] Parrish, A., Chen, A., Nangia, N., Padmakumar, V., Phang, J., Thompson, J., ... & Bowman, S. R. (2021). BBQ: A hand-built bias benchmark for question answering. arXiv preprint arXiv:2110.08193.

---

> ### Author Response · Authors · 2024-11-16
>
> Reviewer comment
> ---
>
> W2. The observations are interesting, but the explanations for the observations are not sufficient.
>
> ---
> Response
> ---
>
> We would love for the opportunity to clarify anything that is unclear. For us to take targeted steps, can you please elaborate on what specific observations you are referring to in this statement, and why you think they are not sufficient?

---

> ### Author Response · Authors · 2024-11-16
>
> Reviewer comment
> ---
>
> W3. The novelty is quite limited. It neither does not provide deep explanations for the observations nor any methods to mitigate the bias.
>
>
> ---
> Response
> ---
>
> The primary purpose of this research is to understand if intrinsic bias in pre-trained models can transfer to downstream tasks upon prompting, to gain fundamental understanding into this adaptation strategy that millions of real-world users leverage to interact with causal LLMs regularly [1]. Using consistent and compatible metrics to evaluate intrinsic and extrinsic biases, careful experimental setup with systematic variation to prompt contexts, we establish that intrinsic gender biases in causal models do transfer to downstream tasks when adapted using prompts. This finding is in contradiction to existing literature in bias transfer analysis, which primarily studies MLMs and fine-tuning adaptations, amounting to a critical novel finding regarding the bias impacts of a widely used adaptation technique.
>
> We believe this finding will be foundational to increasing awareness for, and incentivize important future research into (1) bias impacts of various adaptation strategies, (2) mechanistic explanations for why bias transfers, now that we have established that bias does transfer, and (3) targeted mitigations for fairer pre-training of models that will be prompted to achieve downstream tasks that goes beyond just reducing absolute bias numbers, but removes bias correlation all together. Given that our research uncovers a key novel finding that bias does transfer from pre-trained causal models to downstream tasks using prompting, we believe ICLR to be an appropriate venue for this research given its guidelines for significance of research:
>
> 1. We encourage reviewers to be broad in their definitions of originality and significance.
>
> 2. Submissions bring value to the ICLR community when they convincingly demonstrate new, relevant, impactful knowledge.
>
>
> We agree with you that finding an explanation about why bias transfers, and even mitigating it beyond what we have shown in the paper (Table 2 demonstrates bias mitigation when “Fair” prompts are used, for instance) would be ideal as an extension to this work. However, this is a complex challenge and cannot be resolved in a single work. Only by raising awareness to the problem can the community can be motivated to work on these challenges and tackle them from different perspectives.
>
> Ultimately, the question is, is it valuable to share our findings that contradict previous literature, revealing that gender biases in pre-trained models indeed transfer to downstream tasks through prompting, a widely used adaptation mechanism?
>
> ---
> References
> ---
>
> [1] Al-Dahle, A. (2024, August 29). With 10x growth since 2023, Llama is the leading engine of AI innovation. ai.meta.com. https://ai.meta.com/blog/llama-usage-doubled-may-through-july-2024/

---

> ### Author Response · Authors · 2024-11-16
>
> Reviewer comment
> ---
>
> Q1. The observations are interesting, but they are kind of superficial for me. Could the authors provide any analysis to explain why the correlations exist? For example, some analysis is based on representations or logit lens.
>
> ---
> Response
> ---
>
> Thank you for bringing this up, we recognize that our explanation of the correlations can be made clearer. In our paper, Pearson correlation is measuring alignment in bias direction between intrinsic and downstream biases upon prompting (i.e., does a pro-stereotypical bias for an occupation remain pro-stereotypical with and without prompt adaptation, and similarly for anti-stereotypical bias), across all occupations and random experimental seeds. Despite significant bias reduction using common prompt-based mitigation techniques (as illustrated using “Fair” prompts in Table 2 for Llama3 8b), a strong correlation persists for models (for instance, line 335 specifies that bias correlation for Llama3 8b is >0.92). We find this is because, although absolute biases decrease upon mitigation, occupational stereotypes maintain consistent directional tendencies for gender-occupation combinations. We additionally find that real-world occupational gender biases, as captured by the US Bureau of Labor Statistics, explain the directional stereotypical tendencies in models. We will illustrate and explain this clearly in the main body of the paper to help better interpret our findings.
>
> As for running a deeper analysis at the representation or logit layer, we believe this is out of scope for this work as we are focused on answering our key question of “can bias transfer in prompt-adapted models”. We do completely agree that this would be an important analysis for future work, and we plan to explicitly highlight this in the paper.

---

> ### Author Response · Authors · 2024-11-16
>
> Reviewer comment
> ---
>
> Q2. If in-context learning (prompting) cannot be the promising solution for bias transfer. What is the next step? Could the authors talk more about the future work?
>
> ---
> Response
> ---
>
> This is a great question! Our finding raises awareness for two new key areas of future research, which we will detail in a new future work section in our paper:
>
> 1. Causal explanations for biases and prompts: Now that this works establishes bias transfer in causal models under prompting, it motivates research into causal or mechanistic explanations for (1) why intrinsic and extrinsic biases in causal models correlate even when subject to variation in pre-prompts and prompt context, (2) how prompts impact models on a representational or intermediate activation level, and (3) why bias-inducing prompts are more effective at altering biases than fairness-inducing prompts.
>
> 2. Fairer pre-trained models: Given our finding that intrinsic biases in causal models transfer to downstream tasks, we recommend mitigating intrinsic biases at the pre-training stage. Our finding incentivizes (an already budding area of) research into bias interventions at various stages of the pre-training pipeline, rather than just addressing it in the adaptation strategy like prior BTH work suggests. More specifically, this encourages research into (1) the challenging problem of careful and responsible data curation / collection for large scale LLM pre-training, (2) fairness aware pre-training optimization strategies, (3) post-training mitigation of intrinsic biases in causal LLMs.

---

> ### Author Response · Authors · 2024-11-25
> **Eager to engage with you during rebuttal**
>
> Thanks you for taking the time to review our submission. We are eager to engage with you during the rebuttal. What are your thoughts on our response and is there anything else we can help clarify?
>
> Looking forward to hearing from you on our paper.

---

> ### Comment · Reviewer_WiC1 · 2024-11-25
>
> Thank you for your response. For your question to W2, to explain the behaviors of LLMs, I believe here are some ways that you can refer to,
>
> 1. **From the representation perspective.** You can analyze the internal representations in any middle layer of the LLMs. For example, what is the difference between prompts with Neutral/Fair/Biased instructions at the representation level? what is the difference between Intrinsic/Zero-shot/Few-shot adaptation at the representation level? Which layer affects bias/unbiased behavior the most?
>
> 2. **From the attention perspective.** Besides internal representations, you can also analyze them from the attention. For example, for bias/natural instructions, which part of the instructions is the LLM mainly focusing on? Will the LLM focus more on some extremely biased words in the biased instructions?
>
>
> Here are some papers about the LLM explainability that might help.
>
> [1] Eliciting Latent Predictions from Transformers with the Tuned Lens
>
> [2] Privileged Bases in the Transformer Residual Stream
>
> [3] On the Role of Attention in Prompt-tuning
>
> [4] Interpreting Bias in Large Language Models: A Feature-Based Approach

---

> ### Comment · Reviewer_WiC1 · 2024-11-25
>
> For your response in W3, I am not saying that your findings are not valuable. Instead, I appreciate your observation that revealing that gender biases in pre-trained models indeed transfer to downstream tasks through prompting. However, my point is that this contribution is still not sufficient to meet the bar of ICLR, and that is why I propose W2.
>
> I appreciate your post on the ICLR's guideline for the significance of the research, but I hold my view according to this guideline that the authors should dive deeper to explain the reasons for the phenomenon they observed. For this question, I am open to any further discussion with authors, ACs, and other reviewers.

---

> > ### Author Response · Authors · 2024-11-29
> > **Updated PDF and new experimental results**
> >
> > We thank you for your thoughtful feedback, which helped clarify and strengthen our paper. We are concurrently also considering other suggestions to further improve our contribution.
> >
> > Specifically:
> >
> > 1. We uploaded a new version of PDF with clarifications highlighted in blue and red.
> >
> > 2. We conducted additional experiments to **offer additional empirical analysis that reveal that bias does transfer in causal models across different demographics (such as age, nationality, socio-economic status and nationality), and for CoT prompting**. These findings significantly amplify the impact of our research, extending its relevance and implications beyond just binary-gender, and traditional zero- and few-shot prompting paradigms. We included our initial results in our detailed comment above in the open review forum, and plan to complete and update the camera ready version of the paper if our work is accepted to ICLR.
> >
> > We kindly request you to consider the provided clarifications and new experimental findings in your assessment of our work.

---

> > > ### Author Response · Authors · 2024-12-02
> > > **[Message 2/2] New supplementary results on attention mechanism**
> > >
> > > **B. On the origin of biases**
> > >
> > > We go beyond simply looking at individual (pronoun, occupation) pairs to study attention differences for (1) groupings of pairs, such as (male stereotypical occupation with male pronoun; different groups presented in rows 1-4 in Table 1), and (2) pairs containing male vs female pronouns (presented in rows 5 and 6 in Table 1). For each data split in Table 1, we compute bias in unambiguous WinoBias sentences as the difference in activation scores between a correct pairing (when the given pronoun is the correct answer) and an incorrect pairing (when the given pronoun is the incorrect answer).
> > >
> > > Our analysis reveals that layers L0 and L8 exhibit the most pronounced differences in attention activation when comparing pro- and anti-stereotypical data splits, as well as male and female pronoun splits, using unambiguous WinoBias data. Notably, despite a mean attention activation of approximately 3.9e-05, we observed attention values in layers L0 and L8 that were significantly larger, often by several orders of magnitude. This disparity indicates a substantial difference in activation within these attention heads. **These findings demonstrate that specific attention heads exhibit biased behavior towards different demographic groups. This bias suggests that targeting these attention heads could be a viable strategy for mitigating bias in language models.**
> > >
> > > | **Occupation - pronoun pair** | **Aggregate attention activation across all heads and layers** |  | **Notable per-layer results** |  |
> > > |:---:|:---:|:---:|:---:|:---:|
> > > |  | **Mean** | **Standard deviation** | **Layer 0** | **Layer 8** |
> > > | Female stereotyped occupation + female pronoun  (pro-stereotypical) | 3.95E-05 | 0.01949 | 0.10101 | 0.0309 |
> > > | Male stereotyped occupation + female pronoun  (anti-stereotypical) | 3.95E-05 | 0.02279 | 0.118 | 0.033 |
> > > | Female stereotyped occupation + male pronoun  (anti-stereotypical) | 3.95E-05 | 0.01886 | 0.095 | 0.039 |
> > > | Male stereotyped occupation + male pronoun  (pro-stereotypical) | 3.95E-05 | 0.026 | 0.13 | 0.053 |
> > > | Female pronouns only | 7.89E-05 | 0.00656 | 0.0176 | 0.0018 |
> > > | Male pronouns only | 7.89E-05 | 0.00786 | 0.03472 | 0.01389 |
> > >
> > > ***Table 1. Difference in activation values between correct pairing (when the given pronoun is the correct answer) and incorrect pairing (when the given pronoun is the incorrect answer), for unambiguous sentence in WinoBias. Among all 32 model layers in Mistral, layers 0 and 8 consistently for producing the largest activation differences under various scenarios.***
> > >
> > > ---
> > >
> > > **C. On intrinsic bias mitigation via attention steering**
> > >
> > > We developed targeted bias mitigation strategies focused on attention heads contributing the highest attention scores across occupation-pronoun pairs. We studied effect of attention steering on bias mitigation by uniformly replacing attention head output values to the mean activation value in attention heads that show most difference in activation for different demographics.
> > >
> > > We find that intervening with the 10 most biased attention heads (10 heads that show the most different in activation for different demographics) produces the best fairness, as observed in Table 2. In this setting, bias decreased (overall Stereotype Bias (SB) reduced from 34% to 27%), mainly driven by improvements for unambiguous sentences (SB reduced from 22.40% to 12.66%), but bias is not completely eliminated. **This finding suggests attention steering as a mechanism that may reduce model biases, and potentially also bias transfer.**
> > >
> > > | Heads updated | Pro-stereo RPA | Anti-stereo RPA | Average RPA | Ambiguous SB | Non-ambiguous SB | Average SB |
> > > |:---:|:---:|:---:|:---:|:---:|:---:|:---:|
> > > | None (baseline) | 95.96% | 73.61% | 83.79% | 45.72% | 22.40% | 34.06% |
> > > | Top 1 head | 97.47% | 79.29% | 88.38% | 44.70% | 18.25% | 31.46% |
> > > | Top 5 heads | **97.73%** | **84.85%** | **91.29%** | 42.78%  | 13.05% | 27.83% |
> > > | Top 10 heads | 96.46% | 84.09% | 90.28% | **41.54%** | **12.66%** | **26.95%** |
> > > | Top 20 heads | 96.09% | 78.91% | 87.50% | 44.78% | 17.19% | 30.97% |
> > >
> > > ***Table 2. Performance (RPA) and fairness (A-SB) of Mistral 3 7B using intrinsic adaptation. RPA is measured on only unambiguous sentences whereas A-SB is measured on all data. The best RPA and SB values are bolded.***
> > >
> > > ---

---

> > ### Author Response · Authors · 2024-12-02
> > **[Message 1/2] New supplementary experiments on attention mechanism**
> >
> > **TL;DR:** Based on your feedback, we conducted an empirical analysis of attention mechanisms to further investigate model biases. While we performed numerous experiments on Mistral 3 7B, we report here the most relevant findings related to bias. Our key observations are as follows:
> >
> > * **On bias transfer:** When comparing intrinsic attention patterns to those observed during prompting, we find that the patterns across attention heads remain strikingly similar, which may corroborate the observed bias transfer. Differences are minimal, involving only two heads in layer L0 and one in layer L16.
> > * **On the origin of biases:** Layers L0 and L8 exhibit the largest differences in attention activation between pro- and anti-stereotypical data splits, as well as male and female pronoun splits, particularly with unambiguous WinoBias data. This suggest that the attention mechanism in these layers may play a role in inducing bias. These layers / attention heads may therefore be promising targets for bias mitigation strategies.
> > * **On intrinsic bias mitigation via attention steering:** By selectively modifying the attention heads that exhibit the largest disparities in attention scores for occupation-pronoun pairs, we observed a notable reduction in intrinsic bias. Although this intervention did not completely eliminate bias, it demonstrates the potential for targeted mitigation strategies to decrease bias in language models.
> >
> > Below, we provide details about the experimental setup and a more comprehensive analysis of the results. If this additional analysis addresses your concerns, we are happy to incorporate it into the manuscript. Thank you again for your constructive feedback and for helping us improve our work.
> >
> > **Setup.** We study biases in the attention heads of Mistral 3 7b instruction fine tuned model. Mistral uses a sliding window attention mechanism, and possesses 32 attention heads across its 32 layers. We present WinoBias sentences formatted as below to models, and capture the model attention outputs. We track attention scores for each combination of occupation and pronoun in the input, analyzing patterns across layers and attention heads. For example, we log scores for pairs like (doctor, her) and (nurse, her). Then, we count how often each attention head assigns the highest score to a pair. Finally, we combine these counts across all occupation-pronoun pairs, and analyze them at varying levels of granularity.
> >
> > * Intrinsic: ```<s> The doctor asked the nurse how her day was.```
> > * Zero-shot: ```<s>[INST] Choose the right option for the question using the context below.\nThe doctor asked the nurse how her day was.```
> >
> > **A. On the bias transfer.** We study the difference in attention values between intrinsic and zero-shot prompting setups. Our analysis reveals a remarkably low variance in the contributing attention heads between intrinsic and zero-shot prompting. Specifically, we observed the following:
> >
> > * The 3 most active attention heads exhibit an absolute difference of merely 0.12 +/- 0.03 between intrinsic and zero-shot prompting.
> > * In contrast, the remaining 1021 attention heads display a negligible absolute difference of 5e-3 +/- 2e-2.
> >
> > Notably, even when examining the difference scores within the 3 most active attention heads, the values are extremely small, indicating minimal variation between intrinsic and zero-shot prompting. **We hypothesize that the high similarity in attention activations under intrinsic and prompting evaluations may indicate why we see strong bias transfer in our paper.** A more detailed visualization of these findings can be seen in the heat maps that will be included in the camera-ready version of our paper (as we are unable to upload figures to open review or update PDF at this time), providing a clearer illustration of the differences in attention head contributions between intrinsic and zero-shot prompting.

---

### Official Review · Reviewer_cLVc · 2024-11-06

**Soundness:** 3
**Presentation:** 3
**Contribution:** 1
**Rating:** 6
**Confidence:** 3

**Summary:**

### Summary

- The paper studies the "Bias Transfer Hypothesis" for modern LLMs.
- The bias transfer hypothesis states that biases during pre-training (intrinsic biases) translate to downstream task specific harms (extrinsic biases) and harms.
- Several previous works have studied this in the context of MLMs but not in the context of modern LLMs.
- The previous works did not find strong evidence in support of the Bias Transfer Hypothesis i.e. a strong causal effect of intrinsic biases on extrinsic biases
- In this work, the authors show that there is indeed a positive correlation between intrinsic and extrinsic biases for modern autoregressive LLMs.
- Furthermore, they experiment with various fairness and bias inducing prompts and tinker with the occupation and bias profile of in-context-examples and find that these have relatively small to no effect on changing the bias direction of the model.
- They use the following metrics in their experiments
	-- Referrent Prediction Accuracy
	-- Occupational Selection Bias
	-- Aggregate Selection Bias
- These metrics are consistent with standard metrics in the literature and are easy to follow. The visualization and plots in the paper are immensely informative and I commend the authors for the simplicity in presentation of their results
- In summary all the different experiments, indicate that Bias Transfer Hypothesis is indeed true for modern LLMs. However, I am a bit unsure about the setting in which this has been evaluated. (See my comments Weaknesses below)

Please consider citing the following additional work in the field of LM fairness and evaluation of bias in LMs
- https://arxiv.org/pdf/2208.01448#page=14.61
- https://aclanthology.org/2022.findings-acl.55.pdf
- https://aclanthology.org/2022.acl-long.401.pdf
- https://ojs.aaai.org/index.php/AAAI/article/view/26279
- https://link.springer.com/chapter/10.1007/978-3-030-62077-6_14

**Strengths:**

- Well written , easy to follow, results mostly anticipated.
- Great plots and visualizations and good ablations.

**Weaknesses:**

- I would like to suggest a point of discussion regarding the paper's conceptual framework. The characterization of prompting as an adaptation technique, while innovative, may benefit from further elaboration. Previous studies primarily examined bias transfer in the context of fine-tuning, which involves direct parameter modification. Given that prompting operates through input manipulation rather than model parameter changes, it represents a distinctly different approach to model behavior modification.
- Second, the results would benefit from additional empirical analysis. While the current findings are valuable, incorporating a broader range of intrinsic and extrinsic bias metrics from the literature could provide more comprehensive insights. This expansion could help establish stronger connections between this work and existing research in the field, and potentially reveal additional patterns in bias transfer mechanisms.

**Questions:**

See Weakness above.

---

> ### Author Response · Authors · 2024-11-16
>
> Reviewer comment
> -------------------------------------------------------------------------------------------------------------------------------------------------------------
>
> Please consider citing the following additional work in the field of LM fairness and evaluation of bias in LMs
>
> https://arxiv.org/pdf/2208.01448#page=14.61
>
> https://aclanthology.org/2022.findings-acl.55.pdf
>
> https://aclanthology.org/2022.acl-long.401.pdf
>
> https://ojs.aaai.org/index.php/AAAI/article/view/26279
>
> https://link.springer.com/chapter/10.1007/978-3-030-62077-6_14
>
> -------------------------------------------------------------------------------------------------------------------------------------------------------------
> Response
> -------------------------------------------------------------------------------------------------------------------------------------------------------------
>
> Thank you for the suggestions. We are happy to enrich our related works section with the above recommendations on evaluation of gender biases in LLMs.

---

> ### Author Response · Authors · 2024-11-16
>
> Reviewer comment
> ----------------------
>
> W1. I would like to suggest a point of discussion regarding the paper's conceptual framework. The characterization of prompting as an adaptation technique, while innovative, may benefit from further elaboration. Previous studies primarily examined bias transfer in the context of fine-tuning, which involves direct parameter modification. Given that prompting operates through input manipulation rather than model parameter changes, it represents a distinctly different approach to model behavior modification.
>
> -------------------------------------------------------------------------------------------------------------------------------------------------------------
> Response
> ----------------------
> This is a valid point. We agree, and as a result will explicitly position prompting as being distinctly different in its interactions with model parameters from fine-tuning. We plan to include the following sentence in the intro after line 83 which motivates the need to study bias transfer under prompt-based adaptations
>
> “Additionally, prompting is distinctly different from fine-tuning strategies that were previously employed to study bias transfer in language models. Fine-tuning involves direct parameter modification, whereas prompting operates through input manipulation, which is an entirely different paradigm of interaction with model where bias transfer is less understood. This further motivates study of bias transfer in causal models under prompting.”
>
> Does this sufficiently address your point?

---

> ### Author Response · Authors · 2024-11-16
>
> Reviewer comment
> ----------------------
> W2. Second, the results would benefit from additional empirical analysis. While the current findings are valuable, incorporating a broader range of intrinsic and extrinsic bias metrics from the literature could provide more comprehensive insights. This expansion could help establish stronger connections between this work and existing research in the field, and potentially reveal additional patterns in bias transfer mechanisms.
>
> ------------------------------------------------------------------------------------------
> Response
> ----------------------
>
> Thank you for the opportunity to clarify. Our primary research question in this work is “can bias transfer from pre-trained causal models upon prompting?” While additional experiments and analysis might further strengthen our conclusions, the experiments and analysis that we present in the paper already confirm our hypothesis that bias does transfer from pre-trained causal models upon prompting. This finding is in contradiction to existing literature in bias transfer analysis, which primarily studies MLMs and fine-tuning adaptations, amounting to a critical fundamental understanding into bias effects of the prompting adaptation that millions of real-world users leverage to interact with causal LLMs regularly [2]. Given that our research uncovers this key novel finding that bias does transfer from pre-trained causal models to downstream tasks using prompting, we believe ICLR to be an appropriate venue for this research given its guidelines for significance of research:
>
> 1. "We encourage reviewers to be broad in their definitions of originality and significance."
>
> 2. "Submissions bring value to the ICLR community when they convincingly demonstrate new, relevant, impactful knowledge."
>
> However, we agree with the reviewer that there is an opportunity to better motivate our choice of metrics in the paper, we intend to clearly include the below rationale for the need for RPA, O-SB and A-SB, and ground this choice in shortcomings of existing metrics.
>
> “Current metrics used to measure intrinsic and extrinsic biases are largely incompatible and inconsistent, as noted in previous works [3, 4], typically an artifact of many previous works using different datasets to separately study intrinsic and extrinsic biases. Therefore, to prevent incorrect interpretations of model behavior and to increase confidence in our analysis, we identified a need to formulate a simple unified metric that can be consistently used to evaluate, both, intrinsic and extrinsic prompt-induced biases. This motivated us to introduce RPA, O-SB and A-SB, which are all used to measure both intrinsic and extrinsic model behaviors”
>
> Although our primary hypothesis is confirmed with the current experimental setup, in lieu of your feedback, we identify that the BBQ-lite [1] evaluation metric used to compute extrinsic models biases can be adapted to also perform intrinsic bias evaluation. Therefore, we are currently running experiments using the BBQ-lite [1] dataset and metric to supplement our current findings with this existing bias metric, which we plan to include in our paper in the upcoming week. Apart from this, if there are other metrics that can be used as a consistent measure across intrinsic and extrinsic biases, and that you think will yield insightful analysis beside ones we already use in the paper, can you please enumerate them?
>
> ------
> References
> -----------
>
> [1] Parrish, A., Chen, A., Nangia, N., Padmakumar, V., Phang, J., Thompson, J., ... & Bowman, S. R. (2021). BBQ: A hand-built bias benchmark for question answering. arXiv preprint arXiv:2110.08193.
>
> [2] Al-Dahle, A. (2024, August 29). With 10x growth since 2023, Llama is the leading engine of AI innovation. ai.meta.com. https://ai.meta.com/blog/llama-usage-doubled-may-through-july-2024/
>
> [3] Delobelle, P., Tokpo, E. K., Calders, T., & Berendt, B. (2022, January). Measuring fairness with biased rulers: A comparative study on bias metrics for pre-trained language models. In Proceedings of the 2022 Conference of the North American Chapter of the Association for Computational Linguistics (pp. 1693-1706). Association for Computational Linguistics.
>
> [4] Cao, Y. T., Pruksachatkun, Y., Chang, K. W., Gupta, R., Kumar, V., Dhamala, J., & Galstyan, A. (2022). On the intrinsic and extrinsic fairness evaluation metrics for contextualized language representations. arXiv preprint arXiv:2203.13928.

---

> > ### Comment · Reviewer_cLVc · 2024-11-17
> > **Response to Author's Comments**
> >
> > Thank you for your response. Given the experimental setup using prompt adaptation, I find these results to be expected. However, I believe the paper could still be valuable, so I have increased my score.

---

> > > ### Author Response · Authors · 2024-11-29
> > > **Updated PDF & new experimental results.**
> > >
> > > We thank you for your thoughtful feedback, which helped clarify and strengthen our paper.
> > >
> > > Specifically:
> > > 1. We uploaded a new version of PDF with clarifications highlighted in blue and red.
> > >
> > > 2. We conducted additional experiments to offer **additional empirical analysis that reveal that bias does transfer in causal models across different demographics (such as age, nationality, socio-economic status and nationality), and for CoT prompting**. These findings significantly amplify the impact of our research, extending its relevance and implications beyond just binary-gender, and traditional zero- and few-shot prompting paradigms. We included our initial results in our detailed comment above in the open review forum, and plan to complete and update the camera ready version of the paper if our work is accepted to ICLR.
> > >
> > > We kindly request you to consider the provided clarifications and new experimental findings in your assessment of our work.

---

### Author Response · Authors · 2024-12-02
**Request to consider our supplementary results and clarifications in final paper assessment**

Thank you for your thoughtful feedback and suggestions during the rebuttal period. We appreciate your engagement and have worked to address your comments.

Over the past two weeks, we have conducted **supplementary experiments to further support the presence of bias transfer under prompting** as already demonstrated in our paper. Our new findings include:

* We expanded our analysis beyond binary gender categories to the BBQ-lite dataset, revealing statistically significant bias transfer effects for age, nationality, physical appearance, and socio-economic status.
* We expanded our analysis beyond zero- and few-shot prompting to Chain-of-Thought prompting and again found statistically significant bias transfer.
* **Our investigation into attention activations revealed significantly different behavior of certain attention heads towards different demographics. Notably, attention activations were remarkably similar in intrinsic and prompted states, suggesting the attention mechanism plays a role in model biases.**

We have also updated our PDF to provide clearer motivation for our research and explicitly outline our additions (highlighted in red and blue).

As you finalize your evaluation of our paper, we respectfully request that you consider the additional experiments and clarifications we have provided in response to your feedback.

---

### Meta-Review · Area_Chair_Ji9e · 2024-12-22

**Metareview:**

The paper investigates the bias transfer hypothesis (BTH) in large language models (LLMs), specifically focusing on causal models under prompt adaptation. The authors aim to determine whether intrinsic biases in pre-trained models persist and influence downstream tasks when adapted using zero-shot and few-shot prompting. Through various experiments, the paper shows that intrinsic biases in popular models like Mistral, Falcon, and Llama are strongly correlated with biases observed during prompt adaptation, even when models are pre-prompted to exhibit fair or biased behavior.


Strengths:
* Generally well-written paper, easy to follow
* **Timely and relevant**. The findings are highly relevant for practitioners who use LLMs in real-world applications.
* Thorough and detailed experiments with interesting findings, convincingly presented in good visualizations
* Various informative ablations
* The introduction of new metrics (RPA, O-SB, and A-SB) to measure both intrinsic and extrinsic biases consistently is a notable contribution, addressing inconsistencies in previous studies.

Weaknesses:
* **Limited Evaluation Scope**: The paper primarily focuses on binary gender bias using the WinoBias dataset, which limits the generalizability of the findings to other demographic biases (e.g., race, age, socioeconomic status). Additional experiments were provided in-comments during the rebuttal phase, but not in the final revision (see next point).
* **Some rebuttal clarifications and additional results excluded from final revision**. Multiple reviewers pointed the narrowness of the WinoBias-only evaluation, which prompted the authors to run additional experiments on an additional dataset (BBQ-lite). However, these results do not seem to have been included in the revised version for some reason. For this reason, and the fact that crucial additional results were provided *only* (late) in the rebuttal phase after requests from reviewers and therefore have not gone through the same level of scrutiny, I am downweighing these additional results (but still taking them into account).
* **Very narrow research question**. An informative back-and-forth between the authors and reviewer 1AiY revealed that the novelty of this paper lies on assuming a very narrowly-defined research question: "how prompting affects the transformation of intrinsic biases into downstream biases (for autoregressive models, under a specific bias metric)". On these grounds, the authors decide to limit their comparison and had originally excluded very related work. While most of that work has been incorporated into the Related Work section, the specificity of the research question is weakness of this paper compared to the ICLR standard.
* **Limited exploration of causes and consequences**: the paper does not investigate in detail why certain prompts are more effective at altering bias than others, leaving an important gap in understanding the mechanism by which prompts mitigate or perpetuate biases.
* **Potential Confounders**: As pointed out by reviewer 1AiY, the pretrained open-source models used in this study have all likely been fine-tuned to mitigate bias, which likely introduces confounders that should be controlled for when studying prompt-based debiasing. This is an important question that is not sufficiently discussed/addressed in this work.

Decisions to accept/reject:
Given that this paper focuses on such a narrow research question, the bar for execution is much higher to rise to the level of ICLR. In particular, one would expect:
1. a more-thorough-than-usual discussion of closely related work, to meticulously delineate the contribution
2. going beyond demonstration of the phenomenon, additionally studying the *causes* and *mechanisms* for the phenomenon
3. discussing and controlling for confounders of the (single) phenomenon studied

Thus, although I quite like this paper and enjoy reading it, I must side with the reviewers who do not find it to be above the bar for ICLR.

**Additional Comments On Reviewer Discussion:**

Both authors and (some) reviewers engaged in a very extensive and fruitful discussion (particularly with reviewer 1AiY). The discussion is too long to cover here in detail, and the format and structure that the authors used for their responses to the reviews makes it even harder to follow, but a broad summary of the individual discussions is:

* Reviewer cLVc raised concerns about the limited empirical analysis and the need for additional bias metrics. The authors responded by introducing new metrics and conducting supplementary experiments using the BBQ-lite dataset, which broadened the scope of the analysis.
* Reviewer WiC1 highlighted the exclusive focus on gender bias and the lack of mechanistic explanations. The authors expanded their analysis to include other demographic biases and provided further bias transfer analysis through the lens of attention heads. However, the reviewer remained unconvinced about the sufficiency of the contributions.
* Reviewer 1AiY pointed out the failure to acknowledge closely related prior work and the lack of novelty. The authors clarified the differences between their work and existing studies, and conducted supplementary experiments to show that bias transfer persists even in non-instruction fine-tuned models. The reviewer increased their score (3->5) but maintained concerns about the study’s novelty and the entanglement of fine-tuning and prompting effects.
* Reviewer XDWp suggested exploring additional prompting techniques and expanding the analysis to other biases. The authors conducted new experiments on chain-of-thought prompting and included biases related to age, nationality, physical appearance, and socioeconomic status. This led to an increased rating from the reviewer.

In coming up with the final decision, I downweighed some of WiC1's concerns given their limited engagement with the authors' response, which leaves uncertainty as to whether their concerns were resolved.

---

### Decision · Program_Chairs · 2025-01-22

Reject